# Phenylacetyl-/Trolox- Amides: Synthesis, Sigma-1, HDAC-6, and Antioxidant Activities

**DOI:** 10.3390/ijms242015295

**Published:** 2023-10-18

**Authors:** Rafael Flores, Shoaib Iqbal, Donald Sikazwe

**Affiliations:** Pharmaceutical Sciences Department, Feik School of Pharmacy, University of the Incarnate Word, San Antonio, TX 78209, USA; raflore7@uiwtx.edu (R.F.); shiqbal@uiwtx.edu (S.I.)

**Keywords:** phenylacetic, Trolox, histone-deacetylase-6, antioxidant, drug-likeness

## Abstract

In search of novel multi-mechanistic approaches for treating Alzheimer’s disease (AD), we have embarked on synthesizing single small molecules for probing contributory roles of the following combined disease targets: sigma-1 (σ-1), class IIb histone deacetylase-6 (HDAC-6), and oxidative stress (OS). Herein, we report the synthesis and partial evaluation of 20 amides (i.e., phenylacetic and Trolox or 6-hydroxy-2,5,7,8-tetramethylchroman-2-carboxylic acid derivatives). Target compounds were conveniently synthesized via amidation by either directly reacting acyl chlorides with amines or condensing acids with amines in the presence of coupling agents 1-[bis(dimethylamino)methylene]-1H-1,2,3-triazolo [4,5-b] pyridinium 3-oxide hexafluorophosphate (HATU) or 1,1′-carbonyldiimidazole (CDI). Overall, this project afforded compound **8** as a promising lead with *σ-1* affinity (Ki = 2.1 μM), *HDAC-6* (IC_50_ = 17 nM), and *antioxidant* (1.92 Trolox antioxidant equivalents or TEs) activities for optimization in ensuing structure–activity relationship (SAR) studies.

## 1. Introduction

Alzheimer’s disease (AD) is a progressive poly-factorial brain pathology in which σ-1, HDAC-6 activity, and oxidative stress (OS) have intertwined or synergistic neurodegenerative contributory roles [1,2,3,4]. We opine that *single* small molecules armed with combo σ-1 stimulatory and HDAC-6 and antioxidant or OS inhibitory properties could offer a novel multi-mechanistic approach for developing anti-neurodegenerative and potentially disease-modifying agents for AD.

Sigma-1 receptors are nascent targets for several central nervous system (CNS) diseases, including AD [1,5]. These receptors are functionally diverse, mitochondria-associated endoplasmic reticulum membrane (MAM) single polypeptides, and highly expressed in cognitive (e.g., hippocampus) areas of the brain. Sigma-1 receptors are being targeted for druggability partly because they modulate multiple cellular processes (e.g., apoptosis, excitotoxicity, inflammation, and oxidative stress) and contribute to neuronal survival [6,7,8,9]. Reportedly, σ-1 agonists seem neuroprotective via inositol triphosphate (IP3)-mediated regulation of Ca^2+^ as well by blocking β-amyloid-induced cell death, they increase N-methyl-D-aspartate (NMDA)-associated synaptic plasticity while reducing Ca^2+^ related neurotoxicity, and modulate oxidative stress in several ways, including the induction of antioxidant enzymes (e.g., quinone oxidoreductase or NQO1 and superoxide dismutase or SOD1) [1,5,6,7]. Sigma-1 agonists also promote hippocampal and prefrontal cortex signaling (e.g., via acetylcholine, serotonin, dopamine, glutamate, norepinephrine, and γ-aminobutyric acid or GABA release) and reportedly normalize β-amyloid-stressed mitochondria [1]. In AD animal models, σ-1 receptor agonists have restored memory and exhibited neuroprotection [5].

HDAC-6 is a class IIb α-tubulin hydrolytic enzyme which removes acetyl moieties from N-acetylated lysine groups of histone and nonhistone proteins (e.g., peroxiredoxin enzymes) [10]. Due to its role as a regulatory protein, the HDAC-6 isoform has attracted much interest for druggability in multiple diseases, including CNS disorders. A sampling of evidence in support of this enzyme’s role in AD includes elevated HDAC-6 protein levels in AD brains (especially in the cortical and hippocampal regions) versus normal aged comparators, diminished tau phosphorylation in both HDAC-6 inhibition and knockdown studies, and cognitive improvement observations in cross studies of HDAC-6 knockdown versus amyloid beta (Aβ)-induced memory deficit models. Taken together, HDAC-6 has a multi-pronged contributory role in neurodegeneration and is a validated druggable target for AD [11].

With regard to OS, in general, excessive radical/nonradical reactive species coupled with diminished brain antioxidant enzyme capacity have been linked to neurodegeneration via a variety of mechanisms (lipid peroxidations, Ca^2+^ excitotoxicity, mitochondrial dysfunction, etc.). Additionally, OS also promotes higher HDAC-6 enzyme expression [2,3]. OS-induced alterations, therefore, often culminate in synapse dysfunction, which correlates well with cognitive declines in AD patients [4]. The aberrantly produced Aβ oligomers also induce reactive oxygen species (ROS) production, leading to neuronal apoptosis [12]. Exemplified by glycogen synthase kinase-3 (GSK3) studies, OS can also promote tau hyperphosphorylation, leading to intraneuronal accumulations of neurofibrillary tangles (NFTs) and eventual neuronal death [12]. Interestingly, both Aβ and hyperphosphorylated tau beget more OS and the cycle continues. Although antioxidants have performed poorly in clinical trials, there is still discovery space for brain penetrant lipophilic ones, which could potentially minimize lipid peroxidation in neurodegenerative diseases [13].

Since our ultimate goal is to develop single molecules capable of activating σ-1 while inhibiting both HDAC-6 and OS, we designed our compounds to incorporate three key pharmacophoric elements (i.e., two lipophilic groups (Cap or LG_1_ and LG_2_) plus a linker) common to both σ-1 agonists [14] and HDAC-6 inhibitors [15] (see Figure 1). 

We also included the phenolic Trolox moiety in most of our molecules because it has lipophilic, radical scavenging, or antioxidant properties [16] and to possibly add to HDC6 Cap diversity. For the linker group, we utilized the amide functionality because of its H-bonding capability. Essentially, the chemical basis of action for σ-1 ligands involves a minimum of two hydrophobic and one H-bonding interactions compared to at least two hydrophobic interactions plus a ZBG (hydroxamic and non-hydroxamic acids) for HDAC-6 inhibitors [17]. The Cap motif acts as a lid and prevents substrates from accessing HDAC’s active site, while the ZBG is involved in active site Zn-chelation and H-bonding with the amino acids. Notably, it seems that modifications in the Cap/linker/ZBG functionalities can influence compound selectivity for HDAC isoforms [18].

The rationale for our design is simply that, since σ-1, HDAC-6, and OS (e.g., mitochondria and lipid peroxidation), are independently potentially druggable targets in neurodegenerative pathologies [1,3,19,20], it is plausible that single compounds designed with the *trio-target* capabilities (i.e., σ-1 activation and HDAC-6 and OS inhibitions) could yield novel mechanistic molecules with advantageous neuroprotective properties for testing in AD models. Evidently, there are other ongoing investigations exploring diversely capped HDAC-6 inhibitors for multi-target (e.g., Janus kinase 2 or JAK2 and heat shock protein 90 or Hsp90) activities as potential synergistic pharmacotherapeutic agents for disparate diseases (e.g., fungal and cancer) [21,22,23]. That said, Figure 2 illustrates the 20 phenylacetyl and Trolox amides designed under our proposed pharmacophore hybrid paradigm.

## 2. Results and Discussion

### 2.1. Synthesis

With the exception of **2, 3, 8, 9**, **11**, and **13**, the remaining compounds are known and reported elsewhere for different applications [24,25,26,27,28,29,30,31,32,33,34,35]. Nonetheless, all 20 compounds were conveniently synthesized in two steps by amidation reactions. Synthesis involved either directly reacting acyl halides with appropriate amines (e.g., Figure 1) or simply condensing carboxylic acids and amines using HATU or CDI (e.g., Figure 2) as coupling agents. The follow-up second step mainly involved ZBG (-COOH, -NH-OH, -NH-NH_2_, etc.) elaborations in certain compounds. Compounds **1**, **5**, **16**–**18** were synthesized via direct acylation of nucleophilic amines. Compounds **2**–**4**, **6**–**15**, **19** and **20**, were obtained by reacting amines with carboxylic acids in the presence of either 1-[bis(dimethylamino)methylene]-1H-1,2,3-triazolo [4,5-b] pyridinium 3-oxide hexafluorophosphate (HATU) or 1,1′-carbonyldiimidazole (CDI) as peptide coupling reagents. Compounds **5**, **8**–**9**, and **18** were prepared by reacting intermediate esters with either hydrazine or hydroxylamine to afford hydrazide and hydroxamic acid derivatives. Carboxylic acid **17** was prepared via base facilitated hydrolysis of the intermediate ester. The Appendix A section contains the NMR data related to all compounds.

### 2.2. Drug-Likeness

Guided by Lipinski’s rule of five, oral drug-like properties depend on the molecule’s lipophilicity, flexibility, size, and electronic nature [42,43]. We utilized open-source *Molsoft* (https://www.molsoft.com/servers.html (accessed on 19 July 2023)) computational predictions to identify the above attributes early on in the drug design stages prior to any chemical synthesis [44]. Molsoft generated the drug-likeness molecular descriptors (MW, H-bond acceptors or HBAs, H-bond donors or HBDs, Log*P*, etc.) indicated in Table 1.

Our target compounds were within the Lipinski drug-like space: MWs < 500), numbers of HBAs (Os + Ns ≤ 10), HBDs (NHs + OHs ≤ 5), and calculated logP values were <5. Low MW small molecules are readily absorbed, diffuse, and are easier to transport across membranes. Additionally, polar surface area (PSA) values for all the molecules were below 90 Å^2^—an indicator of enhanced brain bioavailability [45]. Overall, PSA values < 120 Å^2^ characterize plausible drug absorption and, therefore, bioavailability and are calculated from surface areas occupied by oxygen, nitrogen, and the attached hydrogen atoms. PSA values are also closely related to the compound’s hydrogen bonding capability. Except compounds **5**, **16**, and **18**, the rest of the molecules carried calculated logP values in the range of 2.0–4.5, meaning that they are more capable of crossing biological membranes, including the blood–brain barrier (BBB). Per organic chemistry portal https://www.organic-chemistry.org/prog/peo/ (accessed on 17 July 2023), H_2_O solubility (log*S*) affects drug absorption and distribution, while drug-likeness scores imply the commonness of structural features among target molecules and existing drug molecules. Our target compounds were within the water H_2_O solubility range of −2 to −4 for most marketed drugs. Also, most of our compounds (except **1**, **5**, **16**, and **18**) exhibited positive scores—an indication that they contained most of the usual functionalities found in clinical drugs.

### 2.3. Biological Evaluation

#### 2.3.1. Sigma Receptor Binding Affinities

Sigma-1 and 2 affinity experiments (K_i_ values) were carried out at the National Institute of Mental Health’s Psychoactive Drug Screening Program (NIMH PDSP, UNC Chapel Hill, NC). Assay details (i.e., competitive radioligand displacement conditions, etc.) are available at https://pdsp.unc.edu/pdspweb/content/PDSP%20Protocols%20II%202013-03-28.pdf (accessed on 23 August 2023) [46]. Table 2 below indicates the K_i_ values determined for all compounds.

#### 2.3.2. HDAC Fluorescence Activity Assay

HDAC-6 inhibition assays were conducted by the Reaction Biology Corporation (Malvern, PA, USA) using Howitz conditions [47]. The two-step assay protocol involved 1) incubating the purified HDAC enzyme with a fluorogenic substrate possessing an acetylated lysine side chain and 2) treating the above incubated substrate with a developer to produce a fluorophore. Notably, deacetylation of the substrate sensitizes it for the second step. All compounds were dissolved in DMSO and tested in at least 10-dose IC50 mode with threefold serial dilutions (Table 3). “(*R*)-Trichostatin A or (*R*)-TSA”, a potent (e.g., IC50s between 6 and 38 nM) in vivo/in vitro inhibitor of human HDACs, was used as a comparator [48,49]. Percent enzyme activities (relative to DMSO controls) and IC50 values were calculated using GraphPad Prism 4 program (Figure 3).

#### 2.3.3. ORAC Assay

Compound antioxidant capacities were determined using the Oxygen Radical Activity Capacity (ORAC) Assay Kit (catalog #ab233473, Abcam, Cambridge, MA, USA) according to the manufacturer’s protocol. Briefly, the Trolox standard (0.2 mM, 50:50 *v*/*v*, water: acetone) and each test compound (0.2 mM, 50:50 *v*/*v*, water: acetone) were tested at final concentrations ranging from 2.5 µM to 50 µM in the assay. Fluorescein (150 μL) probe solution and 25 μL of Trolox or each test compound were mixed in each well of a black 96-well microplate and incubated at 37 °C for 30 min. Wells which did not contain any antioxidant standard (Trolox) or test compounds served as blanks. The radical initiator (2,2′-azobis(2-amidinopropane) dihydrochloride (AAPH) solution (80 mg/mL, 25 µL) was then added to each well to start the reaction. Immediately, the microplate was placed in a fluorescence microplate reader (BioTek Synergy HTX) and fluorescence was measured at respective wavelengths of 480 nm (excitation) and 520 nm (emission) at 37 °C. Relative fluorescence was measured in increments of 1 min for 60 min. Fluorescence values of Trolox and test compounds for each well were then plotted against time. Individual net areas under the curve (AUC, determined by subtracting the AUC of the blank) were then plotted against different concentrations of Trolox and test compounds to generate calibration curves. Trolox equivalents (TE) for the test compounds were obtained by directly dividing the slopes of each sample by the slope of the Trolox from its standard curve (Table 4) [50]. All compounds were initially screened for antioxidant activity and the six most active ones (i.e., **3**, **4**, **8**, **12**, **18**, and **20**) were run in triplicates to obtain error bars—this data is plotted in the Appendix A section. Curiously, unlike the phenolic Trolox derivatives (i.e., **3**, **4**, **8**, **12**, and **20**), compound **18**′s antioxidant activity can be attributed to the -NH-OH moiety [51].

### 2.4. Compound ***8*** Modelling

Modelling was undertaken to determine and contrast the docking poses using overlays (Figure 5A,B), binding energy scores or affinities, and two-dimensional amino acid (AA) bonding interactions (Figure 6A–D) of our promising lead compound 8 versus native ligands *Trichostatin A* (TSA) for HDAC-6 and *PD144418* for σ1R. The applicable docking protocol details are described elsewhere [52]. Briefly, energy minimized conformers were converted into Protein Data Bank, Partial Charge (Q), and Atom Type (T) or *pdbqt* format. Crystal structures of HDAC-6 complexed with Trichostatin A and σ1R complexed with PD144418 were obtained from the Protein Data Bank database [https://www.rcsb.org/ (accessed on 5 October 2023), PDB, ID: HDAC-6-5EDU, σ1R-5HK1] and imported into Chimera 1.16 to visualize the binding region of the complexes and identify key amino acid (AA) residues involved.

The most stable HDAC-6/5EDU bound conformers of **8** exhibited free-binding energy or affinity scores (in the range −9.1 to −7.2 kcal/mol) comparable to the reference ligand TSA (−8.1 kcal/mol). Notable H-bonding interactions were between the -NH-OH group and GLY 619 or TYR 782. Other interactions included van der Waals forces: π-stacked (PHE 680, PHE 620) for **8**, π-sigma (SER 568, PHE 680, PHE 620) for TSA, plus conventional dispersion forces displayed in both ligands (i.e., via LEU 749, HIS 651, HIS 611, HIS 610, and HIS 500) added to the stability of lipophilic cap areas. On the other hand, the best docked σ1R/5HK1 bound conformers of 8 scored in the range of −10.9 to −9.3 kcal/mol. These scores were also comparable to the native co-crystallized ligand PD144418 docked at 5HK1 (−10.2 kcal/mol). Compound 8 displayed a polar H-bond interaction between its -NH-OH and ASP119, while PD144418 did not display any H-bonding. The most notable interactions shared by both docked ligands were π–anion interactions (GLU 165 and ASP 119), π–alkyl interactions (ALA 178, MET 86, and LEU 98), π–π T-shaped interactions with PHE 10, and π–sigma interactions (HIS 147 and TYR 96), which contributed to the stability of the cap and lipophilic regions of the ligands.

## 3. Materials and Methods

Reactants and solvents were sourced from Thermo Fisher Scientific and Acros Organics, (Pittsburgh, PA, USA), Millipore Sigma-Aldrich (St. Louis, MO, USA), eMolecules (San Diego, CA, USA) and used as received. All reactions were performed according to literature precedence. Melting points, taken on the Mel-temp capillary apparatus, are reported uncorrected. Teledyne Combiflash Rf flash chromatography fitted with Redisep Rf silica gel cartridges was used for compound purifications. Analytical TLC plates from EM Science (silica Gel 60 F_254_) were used. ^1^H and ^13^C spectra were recorded using CDCl_3_ or DMSO-d_6_ as a solvent on a Bruker 300 MHz spectrometer at ambient probe temperature, unless otherwise indicated. ^1^H and ^13^C chemical shifts are reported versus SiMe_4_ and were determined by reference to the residual ^1^H and ^13^C solvent peaks. Coupling constants (*J*) are reported in hertz (Hz). Characterization data are reported as follows: chemical shift, multiplicity (s = singlet, d = doublet, t = triplet, q = quartet, br = broad, m = multiplet), coupling constants, number of protons, and mass-to-charge ratio. Purities and masses of the 20 compounds were determined by analytical reversed-phase high-performance liquid chromatography/mass spectrometer (HPLC/MS) tandem on a Dionex Ultimate 3000 HPLC system. The HPLC method utilized Nova-Pak C18 Column, 60 Å, 4 µm (4.6 mm × 150 mm), at ambient temperature and a flow rate of 1.0 mL/min., CH_3_CN/H_2_O eluent (containing 0.1% acetic acid); gradient, 5% CH_3_CN to 100% CH_3_CN; 8 min.; and UV detection at 254, 260, and 280 nm. nm. Mass was obtained in positive ion mode using heated electrospray ionization (ISQ-HESI) source. MS conditions: capillary voltage, 3.0000 V; drying gas flow, 0.2 mL/min; vaporizer gas temperature, 350 °C; and gas pressure, 28.8 psig. MS data were acquired with Chromeleon 7.

### Chemistry

***Procedure A—****Acylation amidation* using modified organic “Schotten-Baumann” conditions [36]. Phenyl acetyl chloride (1.0 equiv.) was added dropwise to a solution of the appropriate amine (1.0 equiv.) and *N*, *N*-diisopropylethylamine (DIPEA, 3 equivs.) in 2:1 dichloromethane (DCM)/methanol (MeOH) co-solvent at 0 °C. The ensuing mixture was stirred from 0 °C to room temperature (RT) overnight, under N_2_. The reaction was quenched with H_2_O and extracted with DCM. The pooled organic phase was sequentially washed with Na_2_CO_3_ and brine, dried over Na_2_SO_4_, concentrated by rotary evaporator, and purified by flash chromatography using either hexane/ethyl acetate (7:3) or DCM/MeOH (9:1). Solvent removal from product fractions (per TLC) or recrystallizations from DCM afforded the target compounds. Yields are indicated in the experimental write-up of each product.

***Procedure B—****HATU amidation*. A mixture of Trolox (1.0 eqiv.), the appropriate amine (1.5 eq), and HATU (1.5 equivs.) in dimethylformamide (DMF) was stirred at room temperature (RT) for 10 min under N_2_. DIPEA (2.5 equivs.) was added to the reaction mixture, followed by overnight (18–24 h) stirring. The reaction was diluted with DCM, washed with excess H_2_O, and the combined organic phase was washed with brine then dried over Na_2_SO_4_. Product purification was achieved by flash chromatography using hexane/ethyl acetate (7:3) or DCM/MeOH (9:1) as eluents. Solvent removal/recrystallization of product fractions afforded the desired compounds in the yields indicated in individual experimental write-ups. Reaction solvent amounts are included in each experimental write-up.

***Procedure C—****CDI amidation.* After 1 h of stirring 1′-carbonyldiimidazole (CDI) (1.5 equivs.) and Trolox (1.0 equiv.) in tetrahydrofuran (THF) (10 mL), the appropriate amine (1.5 equivs.) was added. The resulting reaction mixture was further stirred, then stirred overnight (24 h) at RT under N_2_. THF was removed under vacuo, H_2_O was added, and organics were extracted with DCM. The combined organic phase was washed with Na_2_CO_3_ and brine, then dried with Na_2_SO_4_. After solvent removal, the crude product was purified via flash chromatography (Combi-Flash Rf) using hexane/ethyl acetate (7:3) or DCM/MeOH (9:1) eluent options. Collection of the relevant fractions yielded the product amounts illustrated in individual experimental write-ups.

***Procedure D—****Hydrazide synthesis.* Excess hydrazine hydrate (50 equivs.) was added in a drop-wise manner to the appropriate methyl ester intermediate (1 equiv.) in absolute ethanol (EtOH) and the reaction mixture was refluxed overnight (24 h). The precipitate which formed was filtered and purified by successively rinsing the solids with cold (0 °C) H_2_O. Reaction solvents and product yields are reported in respective experimental write-ups.

***Procedure E—****Hydroxamic acid synthesis.* Hydroxylamine hydrochloride (15 equivs.) was freebased by stirring it with NaOH (30 equivs.) in MeOH for 10 min and filtered. The methyl ester intermediate (1 equiv.) was added to the hydroxylamine freebase filtrate and the mixture stirred at RT overnight (18 h). The reaction mixture was concentrated by rotary evaporator, dissolved in H_2_O (10 mL), and the solution was acidified to 4 pH with HCl (5N). The crude hydroxamic acid precipitate which formed was filtered and rinsed with ice-cold H_2_O to afford the product. Reaction solvents and product yields are reported in respective experimental write-ups.

***Procedure F—****Hydrolysis.* Excess NaOH (8 equivs.) was added to the appropriate methyl ester intermediate (1 equiv.) in absolute EtOH/THF (10 mL, 1:1) co-solvent. The reaction mixture was refluxed for 4 h and concentrated under vacuo. Water (10 mL) was added and the solution was acidified to 4 pH with HCl (5N). The solid COOH precipitate which resulted was filtered and purified by rinsing with ice-cold H_2_O. Reaction solvents and product yields are reported in experimental write-ups (e.g., compound **17**).

*Methyl 4-((2-phenylacetamido) methyl) benzoate* (**1**). Phenyl acetyl chloride (1.24 g, 8 mmol) was reacted with methyl 4-(aminomethyl) benzoate hydrochloride (1.6 g, 8 mmol) and DIPEA (1.4 mL, 10 mmol) in DCM (10 mL) and methanol (5 mL) at 0 °C for 1 h and at room temperature for 17 h. Reaction work-up was conducted as indicated in procedure “A”. The desired product was isolated as a white crystalline solid (1.3 g, 58% yield), mp 126–129 °C. ^1^HNMR (300 MHz, CDCl_3_) *δ* (ppm): 7.94 (d, *J* = 8.4 Hz, 2H, 2 CH Ar), 7.24 (m, 7H, 5 CH Ar, 2 CH Ar), 6.11 (s, 1H, NH), 4.45 (d, *J* = 6.0 Hz, 2H, CH_2_), 3.90 (s, 3H, CH_3_), 3.63 (s, 2H, CH_2_). ^13^CNMR (75 MHz, CDCl_3_) *δ* (ppm): 171.18, 166.88, 143.61, 134.83, 129.98, 129.46, 129.28, 129.14, 127.52, 127.30, 52.17, 43.79, 43.22. HPLC/MS *t*_R_ 4.82 min.; *m*/*z* calcd. for (C_17_H_17_NO_3_; M+H) 284.1281 found 284.0370. 

*Methyl 4-((6-hydroxy-2,5,7,8-tetramethylchromane-2-carboxamido) methyl)benzoate* (**2**). 6-Hydroxy-2,5,7,8-tetramethylchroman-2-carboxylic acid (500 mg, 2 mmol) was reacted with methyl 4-(aminomethyl) benzoate hydrochloride (604 mg, 3 mmol) and DIPEA (870 µL, 5 mmol) in DMF (10 mL) at room temperature for 24 h with HATU (1.14 g, 3 mmol) as the coupling reagent. The desired product was isolated, using procedure “B”, as a white crystalline solid (436 mg, 55% yield), mp 155–158 °C. ^1^HNMR (300 MHz, CDCl_3_) *δ* (ppm): 7.88 (d, *J* = 8.3630 Hz, 2H, 2 CH Ar), 7.04 (d, *J* = 8.4 Hz, 2H, 2 CH Ar), 6.79 (m, 1H, OH), 4.69 (s, 1H, NH), δ 4.59 (m, 1H, CH), 4.36 (m, 1H, CH), 3.92 (s, 3H, CH_3_), δ 2.63–2.29 (m, 3H, CH_2_, CH), 2.16 (s, 3H, CH_3_), 2.12 (d, *J* = 4.6 Hz, 6H, 2 CH_3_), 1.97–1.83 (m, 1H, CH), 1.60 (s, 3H, CH_3_). ^13^CNMR (75 MHz, CDCl_3_) *δ* (ppm): 174.59, 166.70, 145.89, 144.32, 143.51, 129.84, 129.11, 126.78, 121.81, 121.72, 119.35, 118.12, 78.52, 52.12, 42.67, 29.70, 24.80, 20.66, 12.23, 11.93, 11.32. HPLC/MS *t*_R_ 5.47 min.; *m*/*z* calcd. for (C_23_H_27_NO_5_; M+H) 398.1962 found 398.0298. 

*N-(Benzo[d]oxazol-2-yl)-6-hydroxy-2,5,7,8-tetramethylchromane-2-carboxamide* (**3**). 6-Hydroxy-2,5,7,8-tetramethylchroman-2-carboxylic acid (500 mg, 2 m mol) was reacted with 2-aminobenzoxazole (400 mg, 3 mmol) and DIPEA (870 µL, 5 mmol) in DMF (10 mL) at room temperature for 24 h under HATU (1.14 g, 3 mmol) coupling conditions. Product purification was conducted as indicated in procedure “B”. The desired product was isolated as a white crystalline solid (175 mg, 24% yield), mp 191–193 °C. ^1^HNMR (300 MHz, CDCl_3_) *δ* (ppm): 9.14 (s, 1H, NH), 7.62 (m, 1H, CH Ar), 7.48 (m, 1H, CH Ar), 7.29 (td, *J* = 7.2 Hz, 2H, 2CH Ar), 4.53 (s, 1H, OH), 2.75–2.69 (t, *J* = 6.6 Hz, 2H, CH_2_), 2.55–2.37 (m, 1H, CH), 2.30 (s, 3H, CH_3_), 2.22(s, 3H, CH_3_), 2.11 (s, 3H, CH_3_), 2.09–1.99 (m, 1H, CH), 1.67 (s, 3H, CH_3_). ^13^CNMR (75 MHz, CDCl_3_) *δ* (ppm): 171.64, 153.77, 148.18, 146.23, 143.58, 140.64, 124.79, 124.04, 122.14, 121.82, 119.16, 119.12, 117.74, 110.14, 78.65, 29.34, 24.09, 20.27, 12.28, 12.14, 11.33. HPLC/MS *t*_R_ 5.53 min.; *m*/*z* calcd. for (C_21_H_22_N_2_O_4_; M+H) 367.1653 found 366.9974. 

*6-Hydroxy-2,5,7,8-tetramethyl-N-(4-sulfamoylbenzyl) chromane-2-carboxamide* (**4**). 6-Hydroxy-2,5,7,8-tetramethylchroman-2-carboxylic acid (500 mg, 2 mmol) was reacted with 4-(aminomethyl)benzene sulfonamide (600 mg, 3 mmol) and DIPEA (870 µL, 5 mmol) in DMF (10mL) at room temperature for 18 h under HATU (1.14 g, mmol) conditions. Work-up procedure “B” afforded the desired product as a white powder (420 mg, 50% yield), mp 200–203 °C. ^1^HNMR (300 MHz, DMSO-d_6_) *δ* (ppm): 7.98 (m, 1H, OH), 7.65 (d, *J* = 8.3 Hz, 2H, 2 CH Ar), 7.56 (s, 1H, NH), 7.29 (s, 2H, NH2), 7.13 (d, *J* = 8.3 Hz, 2H, 2CH Ar), 4.42 (m, 1H, CH), 4.25 (m, 1H, CH), 2.61–2.37 (m, 2H, CH_2_), 2.33–2.20 (m, 1H, CH), 2.12 (d, *J* = 10.6 Hz, 6H, 2 CH_3_), 2.03 (s, 3H, CH_3_), 1.82–1.66 (m, 1H, CH), 1.44 (s, 3H, CH_3_). ^13^CNMR (75 MHz, DMSO-d_6_) *δ* (ppm): 174.08, 146.43, 144.54, 144.34, 142.82, 127.17, 125.96, 123.19, 121.86, 120.75, 117.64, 77.85, 42.14, 30.08, 24.91, 20.75, 13.27, 12.63, 12.27. HPLC/MS *t*_R_ 4.62 min.; *m*/*z* calcd. for (C_21_H_26_N_2_O_5_S; M+H) 419.1635 found 418.9639. 

*N-(4-(Hydrazine carbonyl) benzyl)-2-phenylacetamide* (**5**). Compound **1** (400 mg, 1.4 mmol) was treated with hydrazine hydrate (2.5 mL, 50 mmol) in ethanol (10 mL). The reaction mixture was refluxed for 24 h, and procedure “D” work-up afforded the desired product as a white powder (152 mg, 38% yield), mp 163–166 °C. ^1^HNMR (300 MHz, DMSO-d_6_) *δ* (ppm): 9.72 (s, 1H, NH), 8.60 (m, 1H, NH), 7.74 (d, *J* = 8.1 Hz, 2H, 2 CH Ar), 7.29 (m, 7H, 2 CH Ar, 5 CH Ar), 4.48 (s, 2H, NH_2_), 4.30 (d, *J* = 5.8 Hz, 2H, CH_2_), 3.50 (s, 2H, CH_2_). ^13^CNMR (75 MHz, DMSO-d_6_) *δ* (ppm): 170.70, 166.17, 143.11, 136.81, 129.48, 128.69, 127.43, 127.40, 126.85, 42.83, 42.38. HPLC/MS *t*_R_ 3.37 min.; *m*/*z* calcd. for (C_16_H_17_N_3_O_2_; M+H) 284.1394 found 284.0377. 

*N-(1H-benzo[d]imidazol-2-yl)-6-hydroxy-2,5,7,8-tetramethylchromane-2-carboxamide* (**6**). 6-Hydroxy-2,5,7,8-tetramethylchroman-2-carboxylic acid (500 mg, 2 mmol) was reacted with 2-aminobenzimidazole (400 mg, 3 mmol) and DIPEA (870 µL, 5 mmol) in DMF (10 mL) at room temperature using HATU (1.14 g, 3 mmol) conditions. The desired product was isolated, via procedure “B”, as a white crystalline solid (325 mg, 45% yield), mp 236–239 °C. ^1^HNMR (300 MHz, DMSO-d_6_) *δ* (ppm): 12.16 (s, 1H, NH), 10.82 (s, 1H, NH), 7.51 (s, 1H, OH), 7.40 (m, 2H, 2 CH Ar), 7.08 (m, 2H, 2 CH Ar), 2.70–2.44 (m, 3H, CH_2,_ CH), 2.18 (s, 3H, CH_3_), 2.08 (s, 3H, CH_3_), 1.98 (s, 3H, CH_3_), 1.93–1.76 (m, 1H, CH), 1.62 (s, 3H, CH_3_). ^13^CNMR (75 MHz, DMSO-d_6_) *δ* (ppm): 162.77, 146.44, 123.24, 121.68, 120.74, 117.23, 77.88, 36.24, 31.24, 29.86, 24.88, 20.74, 13.22, 12.50, 12.21. HPLC/MS *t*_R_ 5.23 min.; *m*/*z* calcd. for (C_21_H_23_N_3_O_3_; M+H) 366.1812 found 366.0058. 

*6-Hydroxy-N-(isoquinolin-3-yl)-2,5,7,8-tetramethylchromane-2-carboxamide* (**7**). 6-Hydroxy-2,5,7,8-tetramethylchroman-2-carboxylic acid (500 mg, 2 mmol) was reacted with 3-aminoisoquinoline (432 mg, 3 mmol) and DIPEA (870 µL, 5 mmol) in DMF (10 mL) at room temperature for 24 h under HATU (1.14 g, 3 mmol) conditions. The desired product was obtained, using procedure “B”, as a white crystalline solid (223 mg, 30% yield), mp 210–215 °C. ^1^HNMR (300 MHz, DMSO-d_6_) *δ* (ppm): 9.13 (d, *J* = 13.9 Hz, 2H), 8.47 (s, 1H, NH), 8.04 (d, *J* = 8.1 Hz, 1H), 7.92 (d, *J* = 8.2 Hz, 1H,), 7.72 (t, *J* = 7.1 Hz, 1H), 7.62 (s, 1H, OH), 7.54 (t, *J* = 7.1 Hz, 1H), 2.71–2.47 (m, 3H), 2.24 (s, 3H), 2.11 (s, 3H), 2.00 (s, 3H), 1.95–1.80 (m, 1H), 1.59 (s, 3H). ^13^CNMR (75 MHz, DMSO-d_6_) *δ* (ppm): 172.75, 162.76, 152.11, 146.81, 146.05, 143.70, 137.48, 131.60, 128.12, 126.90, 126.61, 126.52, 123.48, 121.41, 121.10, 117.58, 107.18, 78.22, 54.04, 38.70, 36.23, 31.22, 29.41, 24.93, 20.51, 18.52, 17.17, 13.26, 12.43, 12.27. HPLC/MS *t*_R_ 6.58 min.; *m*/*z* calcd. for (C_23_H_24_N_2_O_3_; M+H) 377.1860 found 377.0071. 

*6-Hydroxy-N-(4-(hydroxycarbamoyl)benzyl)-2,5,7,8-tetramethylchromane-2-carboxamide* (**8**). Compound **2** (300 mg, 0.75 mmol) was reacted with hydroxylamine HCl (1 g, 15 mmol) and NaOH (1.2 g, 30 mmol) in MeOH (8 mL) for 18 h. The desired product was obtained using procedure “E” as a white crystalline powder (97 mg, 32% yield), mp 176–179 °C. ^1^HNMR (300 MHz, DMSO-d_6_) *δ* (ppm): 9.15 (br, 2H, NHOH), 7.91 (m, 1H, NH), 7.57 (d, *J* = 8.0 Hz, 2H, 2 CH Ar), 7.01 (d, *J* = 8.0 Hz, 2H, 2 CH Ar), 4.37 (m, 1H, CH), 4.24 (m, 1H, CH), 2.40 (m, 2H, CH_2_), 2.26 (m, 1H, CH), 2.10 (d, *J* = 9.0 Hz, 6H, 2 CH_3_), 2.02 (s, 3H, CH_3_), 1.73 (m, 1H, CH), 1.44 (s, 3H, CH_3_). ^13^CNMR (75 MHz, DMSO-d_6_) *δ* (ppm): 174.02, 164.46, 146.44, 144.53, 143.04, 131.99, 127.09, 126.63, 123.25, 121.80, 120.83, 117.63, 77.86, 55.38, 42.18, 30.09, 24.95, 20.75, 13.28, 12.61, 12.28. HPLC/MS *t*_R_ 4.11 min.; *m*/*z* calcd. for (C_22_H_26_N_2_O_5_; M+H) 399.1915 found 399.0142.

*N-(4-(Hydrazinecarbonyl)benzyl)-6-hydroxy-2,5,7,8-tetramethylchromane-2-carboxamide* (**9**). Compound **2** (400 mg, 1 mmol) was refluxed with hydrazine hydrate (2.5 mL) (50 mmol) in EtOH (10 mL) for 24 h. Per procedure “D”, the target compound was filtered as a precipitate and isolated as white crystalline powder (177 mg, 44% yield), mp 221–223 °C. ^1^HNMR (300 MHz, DMSO-d_6_) *δ* (ppm): 9.69 (s, 1H, NH), 7.93 (m, 1H, NH), 7.66 (d, 2H, *J* = 8.3 Hz, 2 CH Ar), 7.57 (s, 1H, OH), 7.02 (d, 2H, *J* = 8.2 Hz, 2 CH Ar), 4.46 (s, 2H, NH_2_), 4.37 (m, 1H, CH), 4.22 (m, 1H, CH), 2.61–2.36 (m, 2H, CH_2_), 2.31–2.21 (m, 1H, CH), 2.11(d, *J* = 9.5 Hz, 6H, 2 CH_3_), 2.03 (s, 3H, CH_3_), 1.79–1.62 (s, 1H, CH) δ 1.4400 (s, 3H, CH_3_). ^13^CNMR (75 MHz, DMSO-d_6_) *δ* (ppm): 174.024, 166.24, 146.40, 144.54, 143.34, 132.04, 127.27, 126.64, 123.21, 121.81, 120.80, 117.63, 77.86, 42.16, 30.09, 24.94, 20.75, 13.28, 12.62, 12.28. HPLC/MS *t*_R_ 3.95 min.; *m*/*z* calcd. for (C_22_H_27_N_3_O_4_; M+H) 398.2075 found 398.0147. 

*6-Hydroxy-2,5,7,8-tetramethyl-N-(pyrazin-2-yl) chromane-2-carboxamide* (**10**). 6-Hydroxy-2,5,7,8-tetramethylchroman-2-carboxylic acid (500 mg, 2 mmol) was treated with 2-aminopyrazine (300 mg, 3 mmol) in THF (10 mL) in the presence of CDI (487 mg, 3 mmol) as the coupling reagent. The reaction mixture was refluxed for 18 h and processed according to general procedure 3. The desired product was isolated as a white solid (58 mg, 10% yield), mp 140–143 °C. ^1^HNMR (300 MHz, CDCl_3_) *δ* (ppm): 9.48 (s, 1H, CH Ar), 8.78 (s, 1H, NH), 8.26 (s, 1H, CH Ar), 8.16 (s, 1H, CH Ar), 4.48 (s, 1H, OH), 2.69–2.52 (m, 2H, CH_2_), 2.47–2.31 (m, 1H, CH), 2.22 (s, 3H, CH_3_), 2.12 (s, 3H, CH_3_), 2.02 (s, 3H, CH_3_), 1.97–1.87 (m, 1H, CH), 1.55 (s, 3H, CH_3_). ^13^CNMR (75 MHz, CDCl_3_) *δ* (ppm): 173.38, 147.66, 146.02, 143.82, 142.22, 140.44, 136.76, 122.28, 121.71, 119.03, 117.67, 78.27, 29.32, 24.16, 20.36, 12.30, 12.14, 11.37. HPLC/MS *t*_R_ 5.40 min.; *m*/*z* calcd. for (C_18_H_21_N_3_O_3_; M+H) 328.1656 found 328.0040. 

*N-(4,5-Dimethylthiazol-2-yl)-6-hydroxy-2,5,7,8-tetramethylchromane-2-carboxamide* (**11**). A mixture of 6-Hydroxy-2,5,7,8-tetramethylchroman-2-carboxylic acid (500 mg, 2 mmol), 2-amino-4,5-dimethylthiazole (500 mg, 3 mmol), and DIPEA (870 µL) (5 mmol) in DMF (10 mL) was stirred at room temperature for 24 h in the presence of HATU (1.14 g, 3 mmol). Following procedure “B” reaction work-up, the product was isolated as a pale crystalline solid (160 mg, 22% yield), mp 243–246 °C. ^1^HNMR (300 MHz, DMSO-d_6_) *δ* (ppm): 11.18 (s, 1H, NH), 7.53 (s, 1H, OH), 2.61–2.33 (m, 3H, CH_2_, CH), 2.21 (s, 3H, CH_3_), 2.14 (d, *J* = 4.0, 6H, 2 CH_3_), 2.07 (s, 3H, CH_3_), 1.97 (s, 3H, CH_3_), 1.87–1.72 (m, 1H, CH), 1.55 (s, 3H, CH_3_). ^13^CNMR (75 MHz, DMSO-d_6_) *δ* (ppm): 172.16, 153.56, 146.47, 144.48, 142.22, 123.22, 121.76, 120.66, 119.59, 117.19, 77.48, 29.63, 24.73, 20.60, 14.65, 13.23, 12.54, 12.22, 10.75. HPLC/MS *t*_R_ 6.14 min; *m*/*z* calcd. for (C_19_H_24_N_2_O_3_S; M+H) 361.1581 found 360.9848. 

*6-Hydroxy-N-(1H-imidazol-2-yl)-2,5,7,8-tetramethylchromane-2-carboxamide* (**12**). A mixture of 6-hydroxy-2,5,7,8-tetramethylchroman-2-carboxylic acid (500 mg, 2 mmol), 4-aminoimidazole (400 mg, 3 mmol), DIPEA (870 µL) (5 mmol), and HATU (1.14 g) (3 mmol) in DMF (10 mL) was stirred at room temperature for 24 h and processed using procedure “B”. The product was collected as a white powder (401 mg, 63% yield), mp 224–227 °C. ^1^HNMR (300 MHz, DMSO-d_6_) *δ* (ppm): 11.63 (br, 1H, NH), 10.22 (br, 1H, NH), 7.51 (s, 1H, OH), 6.72 (s, 2H, 2 CH Ar), 2.69–2.37 (m, 3H, CH_2_, CH), 2.14 (s, 3H, CH_3_), 2.08 (s, 3H, CH_3_), 1.99 (s, 3H, CH_3_), 1.90–1.72 (m, 1H, CH), 1.56 (s, 3H, CH_3_). ^13^CNMR (75 MHz, DMSO-d_6_) *δ* (ppm): 172.68, 146.45, 144.59, 140.59, 123.23, 121.68, 120.74, 117.32, 77.64, 29.77, 24.86, 20.67, 13.21, 12.48, 12.23. HPLC/MS *t*_R_ 3.30 min.; *m*/*z* calcd. For (C_17_H_21_N_3_O_3_; M+H) 316.1656 found 315.9852. 

*6-Hydroxy-2,5,7,8-tetramethyl-N-(5-methylisoxazol-3-yl)chromane-2-carboxamide* (**13**). 6-Hydroxy-2,5,7,8-tetramethylchroman-2-carboxylic acid (500 mg, 2 mmol), 3-amino-5-methylisoxazole (300 mg, 3 mmol), DIPEA (870 µL, 5 mmol), and HATU (1.14 g, 3 mmol) in DMF (10mL) was stirred at room temperature for 18 h and worked up according to procedure “B”. The target product was obtained as a white crystalline powder (70 mg, 11% yield), mp 147–150 °C. ^1^HNMR (300 MHz, CDCl_3_) *δ* (ppm): 8.71 (s, 1H, NH), 6.64(s, 1H, CH Ar), 4.37 (s, 1H, OH), 2.68–2.45 (m, 2H, CH_2_), 2.40–2.24 (m, 1H, CH), 2.32 (s, 3H, CH_3_), 2.15 (s, 3H, CH_3_), 2.10 (s, 3H, CH_3_), 2.01 (s, 3H, CH_3_), 1.96–1.83 (m, 1H, CH), 1.50 (s, 3H, CH_3_). ^13^CNMR (75 MHz, CDCl_3_) *δ* (ppm): 173.00, 170.18, 157.35, 146.01, 143.75, 122.25, 121.70, 118.95, 117.63, 96.14, 78.31, 29.44, 24.00, 20.31, 12.64, 12.23, 12.12, 11.31. HPLC/MS *t*_R_ 5.59 min; *m*/*z* calcd. for (C_18_H_22_N_2_O_4_; M+H) 331.1653 found 330.9811. 

*(6-Hydroxy-2,5,7,8-tetramethylchroman-2-yl)(4-(pyrimidin-2-yl)piperazin-1-yl)methanone* (**14**). Following procedure “B”, 6-hydroxy-2,5,7,8-tetramethylchroman-2-carboxylic acid (500 mg, 2 mmol) was reacted with 1-(2-pyrimidinyl) piperazine (492 mg, 3 mmol) and DIPEA (870 µL, 5 mmol) in DMF 10 (mL) at room temperature for 18 h with HATU (1.14 g, 3 mmol) as the coupling reagent. The desired product was isolated as a clear crystalline solid (350 mg, 44% yield), mp 144–147 °C. ^1^HNMR (300 MHz, CDCl_3_) *δ* (ppm): 8.23 (d, *J* = 4.8 Hz, 2H, 2 CH Ar), 6.45 (t, *J* = 4.7 Hz, 1H, CH Ar), 4.28 (s, 1H, OH), 4.23–3.33 (br m, 8H, 4 CH_2_), 2.80–2.65 (m, 1H, CH), 2.61–2.44 (m, 2H, CH_2_), 2.10 (d, *J* = 6.1 Hz, 6H, 2 CH_3_), 2.01 (s, 3H, CH_3_), 1.73–1.59 (m, 1H, CH), 1.55 (s, 3H, CH_3_). ^13^CNMR (75 MHz, CDCl_3_) *δ* (ppm): 171.98, 161.47, 157.74, 145.54, 144.51, 121.56, 121.47, 119.23, 117.99, 110.27, 79.06, 45.99, 44.14, 43.64, 43.15, 38.61, 31.48, 25.35, 21.12, 12.28, 12.15, 11.30. HPLC/MS *t*_R_ 5.52 min; *m*/*z* calcd. for (C_22_H_28_N_4_O_3_; M+H) 397.2234 found 397.0725. 

*N-(Benzo[d][1,3]dioxol-5-ylmethyl)-6-hydroxy-2,5,7,8-tetramethylchromane-2-carboxamide* (**15**). Using procedure “C”, 6-hydroxy-2,5,7,8-tetramethylchroman-2-carboxylic acid (250 mg, 1 mmol) was reacted with piperonylamine (220 mg, 1.5 mmol) in THF (10 mL) at room temperature for 24 h. CDI (243 mg, 1.5 mmol) was added as the coupling reagent. The desired product was isolated as a white powder (40 mg, 10% yield), mp 163–166 °C. ^1^HNMR (300 MHz, CDCl_3_) *δ* (ppm): 6.57 (m, 2H, NH, CH Ar), 6.37 (m, 2H, 2 CH Ar), 5.84 (s, 2H, CH_2_), 4.48 (s, 1H, OH), 4.31 (m, 1H, CH), 4.17–4.00 (m, 1H, CH), 2.62–2.27 (m, 3H, CH_3_), 2.07 (s, 3H, CH_3_), 2.02 (d, *J* = 4.1 Hz, 6H, CH_3_), 1.87–1.75 (m, 1H, CH), 1.48 (s, 3H, CH_3_). ^13^CNMR (75 MHz, CDCl_3_) *δ* (ppm): 174.32, 147.82, 146.74, 145.77, 144.38, 132.07, 121.84, 121.69, 120.24, 119.33, 118.14, 108.14, 107.57, 101.01, 78.46, 42.79, 29.68, 24.75, 20.68, 14.22, 12.24, 11.94, 11.31. HPLC/MS *t*_R_ 5.48 min; *m*/*z* calcd. for (C_22_H_25_NO_5_; M+H) 384.1806 found 384.0279. 

*2-Phenyl-N-(4-sulfamoylbenzyl) acetamide* (**16**). Phenyl acetyl chloride (340 µL, 2.6 mmol) was added to a solution of 4-(aminomethyl)benzenesulfonamine (480 mg, 2.2 mmol) and DIPEA (478 µL, 2.5 mmol) at 0 °C in DCM (10 mL) and stirred from 0 °C to RT for 23 h. The desired product was obtained via procedure “A” as a white crystalline solid (83 mg, 13% yield), mp 200–204 °C. ^1^HNMR (300 MHz, DMSO-d_6_) *δ* (ppm): 8.70 (s, 1H, NH), 7.74(d, *J* = 8.3 Hz, 2H, 2 CH Ar), 7.39 (d, *J* = 8.3 Hz, 2H, 2 CH Ar), 7.27 (m, 7H, NH_2_, 5 CH Ar), 4.33 (d, *J* = 5.9 Hz, 2H, CH_2_), 3.50 (s, 2H, CH_2_). ^13^CNMR (75 MHz, DMSO-d_6_) *δ* (ppm): 170.88, 144.08, 143.02, 136.71, 129.49, 128.72, 127.92, 126.90, 126.12, 42.80, 42.28. HPLC/MS *t*_R_ 3.99 min.; *m*/*z* calcd. for (C_15_H_16_N_2_O_3_S; M+H) 305.0955 found 304.9205. 

*4-((2-Phenylacetamido) methyl) benzoic acid* (**17**). The mixture of compound **1** (400 mg, 1.5 mmol) and NaOH (500 mg, 12.5 mmol) in MeOH (5 mL)/THF (8 mL) was refluxed for 4 h, worked up according to procedure “F”, and the product was isolated as a pale white solid (246 mg, 61% yield), mp 219–222 °C. ^1^HNMR (300 MHz, DMSO-d_6_) *δ* (ppm): 12.86 (s, 1H, OH), 8.63 (s, 1H, NH), 7.87 (d, *J* = 8.2 Hz, 2H, 2 CH Ar), 7.28 (m, 7H, 2 CH Ar, 5 CH Ar), 4.34 (d, *J* = 5.9 Hz, 2H, CH_2_), 3.50 (s, 2H, CH_2_). ^13^CNMR (75 MHz, DMSO-d_6_) *δ* (ppm): 170.86, 167.66, 145.13, 136.73, 129.82, 129.74, 129.48, 128.71, 127.62, 126.89, 42.81, 42.44. HPLC/MS *t*_R_ 4.14 min; *m*/*z* calcd. for (C_16_H_15_NO_3_; M+H) 270.1125 found 269.9982. 

*N-Hydroxy-4-((2-phenylacetamido)methyl)benzamide* (**18**). Compound **1** (400 mg, 1.5 mmol) was added to hydroxylamine HCl (1 g, 15 mmol) and NaOH (1.2 g, 30 mmol) in MeOH (10 mL) and the resulting mixture was stirred at RT for 24 h. The reaction was processed using procedure “E” to afford the target compound as a reddish solid (84 mg, 21% yield), mp 150–154 °C. ^1^HNMR (300 MHz, DMSO-d_6_) *δ* (ppm): 11.30 (s, 1H, NH), 9.11 (s, 1H, OH), 8.83 (s, 1H, NH), 7.69 (d, *J* = 7.7 Hz, 2H, 2 CH Ar), 7.29 (m, 7H, 5 CH Ar, 2 CH Ar), 4.31 (m, 2H, CH_2_). ^13^CNMR (75 MHz, DMSO-d_6_) *δ* (ppm): 170.90, 164.62, 143.29, 136.82, 131.58, 129.50, 128.69, 127.45, 127.38, 126.86, 42.79, 42.35, HPLC/MS *t*_R_ 3.48 min; *m*/*z* calcd. for (C_16_H_16_N_2_O_3_; M+H) 285.1234 found 284.9811. 

*6-Hydroxy-2,5,7,8-tetramethyl-N-(2-(pyridin-3-yl) ethyl)chromane-2-carboxamide* (**19**). 6-Hydroxy-2,5,7,8-tetramethylchroman-2-carboxylic acid (500 mg, 2 mmol) was reacted with 3-(2-Aminoethyl)pyridine (300 mg, 3 mmol) in THF (10 mL) for 18 h using CDI (486 mg, 3 mmol) as the coupling reagent. Procedure “C” was applied here and the product was isolated as a white solid (156 mg, 22% yield), mp 130–133 °C. ^1^HNMR (300 MHz, CDCl_3_) *δ* (ppm): 8.34 (m, 1H, CH Ar), 8.20 (m, 1H, CH Ar), 7.21 (m, 1H, CH Ar), 7.04 (m, 1H, CH Ar), 6.41 (m, 1H, NH), 6.27–5.31 (br, 1H, OH), 3.51 (m, 1H, CH), 3.37 (m, 1H, CH), 2.73–2.20 (m, 5H, CH_2_, CH_2_, CH), 2.11 (s, 3H, CH_3_), 2.03 (s, 3H, CH_3_), 1.94 (s, 3H, CH_3_), 1.81–1.65 (m, 1H, CH), 1.40 (s, 3H, CH_3_). ^13^CNMR (75 MHz, CDCl_3_) *δ* (ppm): 174.59, 149.75, 147.63, 145.97, 144.06, 136.23, 134.17, 123.44, 122.55, 121.57, 120.05, 117.73, 78.26, 39.64, 32.92, 29.50, 24.69, 20.50, 12.54, 11.87, 11.62. HPLC/MS *t*_R_ 3.610 min; *m*/*z* calcd. for (C_21_H_26_N_2_O_3_; M+H) 355.2016 found 355.0197. 

*N′-(4-Chlorobenzoyl)-6-hydroxy-2,5,7,8-tetramethylchromane-2-carbohydrazide* (**20**). 6-Hydroxy-2,5,7,8-tetramethylchroman-2-carboxylic acid (500 mg, 2 mmol) was reacted with 4-Chlorobenzohydrazide (370 mg, 3 mmol) in DMF at room temperature for 18 h with HATU (1.14 g, 3 mmol) as the coupling reagent. The reaction was processed according to procedure “B” and the obtained product was a white crystalline powder (376 mg, 47% yield), mp 205–208 °C. ^1^HNMR (300 MHz, DMSO-d_6_) *δ* (ppm): 10.41 (s, 1H, NH), 9.56 (s, 1H, NH), 7.88 (d, *J* = 8.6 Hz, 2H, 2 CH Ar), 7.56 (d, *J* = 9.3 Hz, 3H, 2 CH Ar, OH), 2.65–2.46 (m, 2H, CH_2_), 2.40–2.24 (m, 1H, CH), 2.11 (d, *J* = 14.1 Hz, 6H, 2 CH_3_), 2.02 (s, 3H, CH_3_), 1.84–1.69 (m, 1H, CH), 1.51 (s, 3H, CH_3_). ^13^CNMR (75 MHz, DMSO-d_6_) *δ* (ppm): 173.39, 164.79, 146.33, 144.54, 137.11, 131.71, 129.76, 129.06, 123.16, 121.99, 120.62, 117.51, 77.80, 30.13, 24.97, 20.48, 13.23, 12.64, 12.28. HPLC/MS *t*_R_ 5.21 min; *m*/*z* calcd. for (C_21_H_23_N_2_O_4_Cl; M+H) 403.1373 found 402.9188. 

## 4. Conclusions

Phenylacetamides (**1**, **5**, **16**–**18**) and Trolox amides (**2**–**4**, **6**–**15**, **19**, **20**) with drug-like properties were designed based on common pharmacophoric features of σ-1 and HDAC-6 ligands and synthesized via amidation. Testing the compounds for σ-1/HDAC-6/antioxidant activities led to the following findings: (i) six compounds (**2**, **8**, and **11**–**14**) with σ-1 affinity Ki values of 2.1–8.3 μM, (ii) two compounds (**8** and **18**) possessed HDAC-6 inhibitory activities of 17 and 77 nM, and (iii) six compounds (**3**, **4**, **8**, **12**, **18,** and **20**) exhibited potent antioxidant activities, and iv) compound **8** (MW = 398.18, ClogP = 2.44) exhibited the desired trifecta activities (i.e., σ-1 affinity (Ki = 2.1 μM), HDAC-6 (IC_50_ = 17 nM), and OS inhibition (1.92 TE)) we were seeking. Interestingly, the modelling data also revealed that **8** bound to both σ-1 and HDAC-6 with similar poses and free energies as the native ligands. Compound **8** will, therefore, undergo additional in vitro testing to further profile its selectivity and will be used as a lead in follow-up SAR studies.

## Data Availability

Not applicable.

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
