# Peer review of "Phenylacetyl-/Trolox- Amides: Synthesis, Sigma-1, HDAC-6, and Antioxidant Activities"

_ijms, 2023, doi:10.3390/ijms242015295_

Round 1

Reviewer 1 Report

The authors in manuscript entitled Benzyl- /Trolox- amides: Synthesis, Sigma-1, HDAC-6, and An-tioxidant Activitiesstudied the design and synthesis of twenty amides (i.e., benzylic acid and trolox or 6-hydroxy-2,5,7,8-tetramethylchroman-2-carboxylic acid derivatives). Target compounds were conveniently synthesized via amidation, either by directly coupling acyl chlorides with amines or through condensing acids with amines in the presence of 1-[bis(dime-thylamino) methylene]-1H-1,2,3-triazolo[4,5-b] pyridinium 3-oxide hexafluorophosphate (HATU) or 1,1'-carbonyldiimidazole (CDI), followed by unravelling (in some compounds) the zinc binding functionality (ZBF) required in HDAC-6 ligands.

Strengths of the study:

- The research article has concluded that this exploratory project yielded compound 8 as promising structural lead for single small molecule (MW= 398.18, ClogP = 2.44) possessing trifecta activities [i.e., σ-1 affinity (Ki = 2.1 μM), HDAC-6 (IC50 = 17 nM) and OS inhibition (1.92 TE)] which will be used in the ensuing SAR studies.

Comments:

1.     Need to write abstract clear and crisp.

2.     Need to write introduction clear and structured. There is no clear agenda for experiment.

3.     Need to write method and material portion again with clear and well structured. This is very unclear and not understanding what you did in this study.

4.     Result and discussion portion is very clumsy. Need to write with elaboration and need to discussed with others experiments. Add more references.

5.     There is need to write again the conclusion with clear and straight.

There are some issues with this article, if these issues are going to resolve then the quality of the paper is not suitable for publication.

1)             Introduction is not well defined

2)             Concluding remark is not clear.

3)             There are few typos and English grammar errors.

4)             Every part is not crisp, concise and well structured.

The manuscript needs extensive English editing. The manuscript includes many untrue statements.

Author Response

Reviewer #1 Comments and our Responses.

We thank the reviewer for their time, effort and comments. We feel that the corrected manuscript adequately addresses the shortcomings (e.g., clarity, conciseness, structure, language, untrue statements) identified in the earlier version and meets the reviewer’s approval.

  1. Comment: Need to write abstract clear and crisp.

Response: The abstract has now been re-written for clarity and succinctness. The wording has been reduced so that the reader can readily get to the intent and key findings of the project.  

  1. Comment: Need to write introduction clear and structured. There is no clear agenda for experiment.

Response: The introduction has been re-written and clearly points out why the project was undertaken (i.e., to develop novel multi-target anti-neurodegeneratives for Alzheimer's). The introduction has been sectioned into paragraphs to provide structure to the sigma/HDAC/oxidative stress content background and interconnectedness.   

  1. Comment: Need to write method and material portion again with clear and well structured. This is very unclear and not understanding what you did in this study.

Response: The methods and material section has been re-written to address the above  comments, however, there is precedent in chemistry journal for writing the section in this  manner (i.e., having a section of general procedures all in one area of the manuscript). We hope the revisions made are satisfactory.

  1. Comment: Result and discussion portion is very clumsy. Need to write with elaboration and need to discussed with others experiments. Add more references.

Response: The results and discussion section has been re-organized and to clearly state what experiments were carried out and the findings. Additionally, we have added  a molecular modelling section for compound 8...this aspect was recommended by another reviewer. Also, more references have been added.

  1. Comment: There is need to write again the conclusion with clear and straight.

Response: The conclusion has been re-written for clarity and points out the key findings of the project.

  1. Comments: Quality of English Language (the manuscript needs extensive English editing). The manuscript includes many untrue statements.

Response: Where applicable the wording used manuscript has been corrected for language errors, grammar, and spell checked. The manuscripts has also been revised to remove factual errors (e.g., the title has also been corrected from “Benzyl-“ to “Phenylacetyl-”).

Reviewer 2 Report

The paper entitled “Benzyl- /Trolox- amides: Synthesis, Sigma-1, HDAC-6, and Antioxidant Activiti” describes the synthesis of novel multitarget compounds with synergistic multi-target mechanisms and their biological evaluation as potential anti-Alzheimer agents. Biological targets of the synthesized compounds were found to be sigma-1 receptor and HDAC, moreover the compounds demonstrated to act as antioxidant agents and taken together this features could potentially lead to a multi-target activity for the treatment of Alzheimer's disease. The work could be depicted as routine medicinal chemistry, without any significant novelty in the chemistry part although the biological section shows interesting results, with some of the compounds demonstrating powerful inhibitory activity. I consider major revision overall and I will support publication only if the following key-points will be clarified.

-       HDAC6 has been a protein involved in the development of dual and multi-target inhibitors in recent years and numerous works highlighted how this feature can be exploited in the development of potent therapeutic agents. I suggest the inclusion of these two references relating to the dual design of HDAC6 and Hsp90 and JAK2 inhibitors (J. Med. Chem. 2018, 61, 14, 6056–6074; Current Medicinal Chemistry 29 (9), 1474-1502).

-       A separate paragraph describing the design of the inhibitors should be inserted. First, HDAC6 selectivity is not guided by the zinc binding group but is mostly determined by the choice of the CAP group as you can see here (https://doi.org/10.1002/cmdc.202000934), moreover please use the common acronym ZBG (Zinc Binding Group) instead of ZBF. The design therefore does not contain Trolox structures, please insert and describe the choice of this functionality.

-       Compound 8 turned out to be the most interesting derivative with improved activity against all the target. A molecular docking study need to be included in the work to explain the pose of 8 inside binding cavity of HDAC6 to clarify the most interesting interaction. Also a similar thing with sigma 1 should be done.

-       Reaction schemes need to be improved. Yields are missing. Please add them. Also description of the reagents employed are not good: please add all reagents, solvents, temperature and reaction time in all the arrows, otherwise use "a", "b" etc. and modify the caption of the schemes.

-       Before Materials and Methods, a paragraph containing "Biological evaluation" should be inserted and the SARs need to be explained. A full discussion of the data need to be inserted in the text. Maybe a table containing all biological data for all the compounds could be inserted. Then the separate tables in the material and methods section could be removed.

Author Response

Reviewer #2 Comments and our Responses.

We thank the reviewer for their time, effort and comments. We hope that the corrected manuscript adequately addresses the shortcomings identified in the earlier version. The reviewer’s comments enabled us to strengthen our manuscript and improve its readability.

  1. Comments: HDAC6 has been a protein involved in the development of dual and multi-target inhibitors in recent years and numerous works highlighted how this feature can be exploited in the development of potent therapeutic agents. I suggest the inclusion of these two references relating to the dual design of HDAC6 and Hsp90 and JAK2 inhibitors (J. Med. Chem. 2018, 61, 14, 6056–6074; Current Medicinal Chemistry 29 (9), 1474-1502).

Response:  We thank the reviewer for this suggestion and have since added a total of three more references (including the two above) which exemplify HDAC-6 in multi-target drug design.

  1. Comments: A separate paragraph describing the design of the inhibitors should be inserted.  First, HDAC6 selectivity is not guided by the zinc binding group but is mostly determined by the choice of the CAP group as you can see here (https://doi.org/10.1002/cmdc.202000934), moreover please use the common acronym ZBG (Zinc Binding Group) instead of ZBF. The design therefore does not contain Trolox structures, please insert and describe the choice of this functionality.

Response:  Thank you for the opportunity to correct and add the suggested HDAC information to our manuscript.  We have made the nomenclature (ZBF to ZBG) changes and clarified the HDAC detail in the design section. However, we did not want the focus of this project to shift from multi-target (sigma/hdac/oxidative stress) to HDAC.

  1. Comment: Compound 8 turned out to be the most interesting derivative with improved activity against all the target. A molecular docking study need to be included in the work to explain the pose of 8 inside binding cavity of HDAC6 to clarify the most interesting interaction. Also a similar thing with sigma 1 should be done.

Response: We thank the reviewer for this suggestion. We have since added the modelling of compound 8 to the results and discussion section. The modes/poses and amino acid interactions are also described.

  1. Comments:Reaction schemes need to be improved. Yields are missing. Please add them. Also description of the reagents employed are not good: please add all reagents, solvents, temperature and reaction time in all the arrows, otherwise use "a", "b" etc. and modify the caption of the schemes.

Response: We have modified the reaction scheme captions to “a”, “b”, etc. and numbered the representative compounds in the two schemes. Otherwise, the reaction details and yields are in chemistry section 3.1.

  1. Comments: Before Materials and Methods, a paragraph containing "Biological evaluation" should be inserted and the SARs need to be explained. A full discussion of the data need to be inserted in the text. Maybe a table containing all biological data for all the compounds could be inserted. Then the separate tables in the material and methods section could be removed.

Response:  We thank the reviewer for this suggestion. We have moved the biological evaluation section to the results and discussion section but kept the data sets as presented in the earlier version because there is IJMS literature precedence for reporting data in this manner. Additionally, SAR aspects were not tackled in this manuscript because that topic will be the primary focus of our next set of manuscripts.  In this paper we simply wanted to share our exploratory design attempt towards molecules possessing combined sigma/hdac-6/antioxidant activities.

Reviewer 3 Report

The manuscript “Benzyl- /Trolox- amides: Synthesis, Sigma-1, HDAC-6, and Antioxidant Activities” is devoted to the investigations of a series of amides bearing antioxidant moiety and activator of σ-1 as potent anti-AD drugs. The study includes synthesis, computational and biological investigations. 5 of the studied compounds are new.

In my opinion, this manuscript suits to the scope of IJMS. I recommend that after minor revision, it can be accepted.

Some comments for the authors:

1. «Benzyl- /Trolox- amides: Synthesis, Sigma-1, HDAC-6, and Antioxidant Activities» should be smth like «Phenylacetic- /Trolox- amides: Synthesis, Sigma-1, HDAC-6, and Antioxidant Activities»

2. «With the exception of 2, 3, 8, 9, 13, the remaining compounds are known and reported elsewhere (mostly in patent literature, for disparate applications).» - refs are required.

3. In section 3.1 refs are required for each known compound to provide the reader an opportunity to compare properties of the compound reported in this paper and in earlier studies.

Author Response

Reviewer #3 Comments and our Responses.

We thank the reviewer for their time, effort and comments. The corrected manuscript is much clearer and has incorporated all the changes suggested by the reviewer.

  1. «Benzyl- /Trolox- amides: Synthesis, Sigma-1, HDAC-6, and Antioxidant Activities» should be smth like «Phenylacetic- /Trolox- amides: Synthesis, Sigma-1, HDAC-6, and Antioxidant Activities»

Response: Thank you for catching our huge error in nomenclature. The title has now been changed to “Phenylactyl-.“

  1. Comment. «With the exception of 2, 3, 8, 9, 13, the remaining compounds are known and reported elsewhere (mostly in patent literature, for disparate applications).» - refs are required.

Response: Thank you for this suggestion. The literature references have been added to the manuscript.

  1. Comment. In section 3.1 refs are required for each known compound to provide the reader an opportunity to compare properties of the compound reported in this paper and in earlier studies.

Response: we thank the reviewer for this insight. However, we have checked many IJMS published manuscripts and the format does not seem to allow inclusion of reference numbers after each title compound in the chemistry section.  That said, all the reported compounds are also searchable by SciFinder.

Round 2

Reviewer 1 Report

Acceptable in Revised form

Ok

Reviewer 2 Report

Dear authors,

I think that the actual manuscript is suitable for a successful publication!

Good luck